# Development and Validation of Huffaz ProHealth 1.0^©^: A Module to Improve the Well-Being of Tahfiz Students in Selangor, Malaysia

**DOI:** 10.3390/ijerph19137718

**Published:** 2022-06-23

**Authors:** Ismarulyusda Ishak, Nurul Najwa Abdul Rahim, Noor Idaya Salim, Cut Ulfah Nihayati Sholeha Teuku Husaini, Izzati Jamaludin, Hafidzoh Mohamad Khalid, Nur Adeena Ahmad Lutfi, Sayyidah Syathiroh Afzaruddin, Ahmad Rohi Ghazali, Arimi Fitri Mat Ludin, Nor Malia Abd Warif, Farah Wahida Ibrahim, Faisal Ariffin Othman, Normah Che Din, Dzalani Harun, Hanis Mastura Yahya, Wan Nor Atikah Che Wan Mohd Rozali

**Affiliations:** 1Biomedical Science Program, Faculty of Health Sciences, Universiti Kebangsaan Malaysia, Jalan Raja Muda Abdul Aziz, Kuala Lumpur 50300, Malaysia; ismarul@ukm.edu.my (I.I.); nurulnajwarhim@gmail.com (N.N.A.R.); nooridayasalim@gmail.com (N.I.S.); cutulfahns@gmail.com (C.U.N.S.T.H.); izzatijamaludin.work@gmail.com (I.J.); hafidzoh97khalid@gmail.com (H.M.K.); adeenalutfi@gmail.com (N.A.A.L.); ssyathiroh@gmail.com (S.S.A.); rohi@ukm.edu.my (A.R.G.); malia.warif@ukm.edu.my (N.M.A.W.); farahwahida@ukm.edu.my (F.W.I.); faisal_ariffin@ukm.edu.my (F.A.O.); p110917@siswa.ukm.edu.my (W.N.A.C.W.M.R.); 2Center for Toxicology & Health Risk Studies Faculty of Health Sciences, Universiti Kebangsaan Malaysia, Jalan Raja Muda Abdul Aziz, Kuala Lumpur 50300, Malaysia; 3Center for Healthy Ageing & Wellness, Faculty of Health Sciences, Universiti Kebangsaan Malaysia, Jalan Raja Muda Abdul Aziz, Kuala Lumpur 50300, Malaysia; hanis.yahya@ukm.edu.my; 4Clinical Psychology and Health Behavior Program, Faculty of Health Sciences, Universiti Kebangsaan Malaysia, Jalan Raja Muda Abdul Aziz, Kuala Lumpur 50300, Malaysia; normahched@gmail.com; 5Center for Rehabilitation & Special Needs Studies, Faculty of Health Sciences, Universiti Kebangsaan Malaysia, Jalan Raja Muda Abdul Aziz, Kuala Lumpur 50300, Malaysia; dzalani@ukm.edu.my; 6Occupational Therapy Program, Faculty of Health Sciences, Universiti Kebangsaan Malaysia, Jalan Raja Muda Abdul Aziz, Kuala Lumpur 50300, Malaysia; 7Nutrition Science Program, Faculty of Health Sciences, Universiti Kebangsaan Malaysia, Jalan Raja Muda Abdul Aziz, Kuala Lumpur 50300, Malaysia

**Keywords:** health intervention, module development, physical health, nutritional, psychological well-being, Tahfiz school

## Abstract

The increase in the number of Tahfiz schools in Malaysia indicates a high demand from the community for Tahfiz education. Tahfiz schools provide a Quran memorization syllabus as the main part of the curriculum at the primary and secondary levels. There is no holistic module that combines learning and health for Tahfiz students in Malaysia. Huffaz ProHealth 1.0^©^ Module is a health intervention module developed explicitly for Tahfiz students by a group of researchers from Universiti Kebangsaan Malaysia Kuala Lumpur (UKMKL). The module encompasses educational and motivational components for the Tahfiz students, and acts as a guide for their teachers and staff to improve the students’ health and quality of life. The module consists of three components: physical health, nutritional and psychological well-being. Each component is divided into several units with specific activities. The Sidek Module Development Model (SDMM) was used as a reference model. The study involved two phases: the development (Phase I) and evaluation (Phase II). In Phase I, a needs assessment was conducted among students and teachers from Tahfiz schools to evaluate their knowledge, attitude, and practice on the related topics. Meanwhile, the module’s validity and feasibility were assessed by healthcare experts and Tahfiz teachers in Phase II. The item–content validity index (I–CVI), content validity value, Tool to Evaluate Materials used in Patient Education (TEMPtEd) score, and expert feedback were evaluated. In conclusion, the Huffaz ProHealth 1.0^©^ module was successfully developed and evaluated. Strong validity values were observed in all components, indicating high suitability to be employed at Tahfiz schools.

## 1. Introduction

The importance of learning and memorizing the Quran to enhance the virtue and mindfulness among Muslim communities in Malaysia is increasingly essential [1]. Most parents send their children to Tahfiz schools, hoping that the latter can achieve a balanced secular and religious education. The term Tahfiz originates from the Arabic language, and it is defined as to memorize. It is a process of memorization by reading, listening, and reciting, specifically the verses of the Quran. The National Tahfiz Education Policy (DPTN) through the Department of Islamic Development Malaysia (JAKIM) has been drafted to empower the Tahfiz education system with the aim to produce professional huffaz in the field of arts and humanities as well as science and technology, and skillful huffaz in the industry. With the implementation of this policy, the government’s vision to have a standard and quality Tahfiz education system will soon be materialized.

However, despite the increasing number of Tahfiz schools in Malaysia, the healthcare management of Tahfiz students is still minimal. This below-standard situation is more prominent among the private Tahfiz schools. They are more diverse in their management systems and curricula, including the aspects of student health management [2]. Furthermore, a study also found that there were weaknesses in Tahfiz education due to the lack of strategies, methods, and motivation in learning [3]. Students’ learning is also deeply packed and imbalanced, as they are made to focus most on completing their Quran memorization syllabus within the allotted time.

The health intervention module refers to planning activities that facilitate the direction and modification of behavior to achieve a set goal [4]. Health intervention programs among adolescents involving educational and motivational approaches have improved students’ quality of life and health status [5]. Thus, the Huffaz ProHealth 1.0^©^ Module was developed to ensure that the Tahfiz students are productive in their academic and Islamic studies, and at the same time, are physically, mentally, and emotionally fit. The module covers three main health aspects: physical health, nutrition, and psychological well-being.

Gamification has been identified as the best tool to encourage students to enthusiastically participate in learning activities [6]. The module’s implementation, mainly through game-based and interactive learning, aims to increase the levels of knowledge, attitude, and practice of Tahfiz students with respect to the three aforementioned aspects.

A needs assessment is necessary for any module’s development. It is done to assess the existing knowledge, abilities, interests, and practices in a group of intended participants [7]. Following a development process, content validation is substantially important to evaluate the quality of an instrument. There are six steps to conduct for content validation: preparing content validity questionnaires; identifying expert evaluators; conducting a content validation assessment; having the experts evaluate each item; acquiring a score for each item; and finally, calculating the content validity index (CVI) [8]. Next, the Tool to Evaluate Materials Used in Patient Education (TEMPtED) was used to evaluate the criteria of a learning material [9]. Hence, this study aimed to report the development and evaluation processes, and their findings on the Huffaz ProHealth 1.0^©^ Module.

## 2. Methodology

Our aim was to develop the Huffaz ProHealth 1.0^©^ module as a comprehensive module towards psychological and physical well-being. The study consisted of two phases (Phase I and Phase II) as adapted from the Sidek Module Development Model 2005. Phase I and Phase II involved the necessary steps illustrated in Figure 1.

### 2.1. ETHICS

Permission to conduct the study was granted by the Association of Selangor Al-Quran Tahfiz Institutions (PITAS). This study was also approved by the Research Ethics Committee Universiti Kebangsaan Malaysia (Human) (UKM PPI/111/8/JEP-2018-394). Parental written consent was also obtained and documented before the execution of the study.

### 2.2. PHASE I

#### 2.2.1. Study Design

Phase I was considered as design and development research (DDR). This phase involved a needs assessment analysis through a cross-sectional study in two Tahfiz schools. Each student and teacher were given a questionnaire in the form of an open-ended and closed-ended questions. These questionnaires were developed to assess the knowledge, attitudes, and practices about a healthy diet, physical activity, and psychological well-being of students; and the teachers’ perceptions of their students. The questionnaires were validated by researchers and expert content in a pilot study and distributed to the students and teachers.

Phase I had nine steps to develop the draft module, involving a needs assessment on the Tahfiz students’ psychological and physical well-being. The inputs from parents and teachers were also taken into consideration. Initially, we set goals for the module (Step 1). General goals related to psychological and physical well-being were identified among Tahfiz students to achieve good overall health. Next, a review of many previous studies was carried out to identify the related rationale, theories, concepts, target group, and time allocation to develop the module (Step 2). A needs assessment was conducted to identify the research needs (Step 3). In Step 4, specific objectives were refined and set to determine the contents of the module. The content selection was determined based on the objectives, research needs, literature review, discussions, and references from various sources to develop the units and the module’s components (Step 5). The strategies and appropriate contents (units and components) were selected to ensure the objectives’ success and the module’s goals (Step 6). The mode of contents delivery was chosen in Step 7 (logistic selection). The module’s content delivery, and those knowledge delivery units and components to students must be sufficient to create the final module. Media is also an essential part of instructional design and module delivery. Appropriate media was used in the module to ensure that the content was delivered effectively (Step 8). Finally, in Step 9, all the contents (units and components) were combined to form a complete module. The module was then evaluated in Phase II.

#### 2.2.2. Study Location

This study was conducted among Tahfiz students in their schools that have been registered under the Selangor Tahfiz Institutions Association (PITAS) located in the district of Petaling, Selangor in Malaysia.

#### 2.2.3. Sampling

The sample size was calculated for this needs assessment study by using G*power 3.1.9.4 for one sample case proportion with *p*-value of 0.1, power of 0.8, and effect size of 0.3. The total number required was 20 participants. This study was conducted on 20 Tahfiz students and only 12 teachers. Purposive sampling was employed with inclusion factors such as boys aged 13 years to 17 years, Tahfiz students from boarding schools, and use of the Malay language for conversation. Exclusion factors included students who were unhealthy, diagnosed with chronic diseases, and using prescribed drugs. The sampling for teachers was universal sampling; all teachers with at least three years of teaching experience in the selected schools were included.

#### 2.2.4. Procedure

Upon the completion of the needs assessment analysis, all researchers, together with the Tahfiz school teachers and administrators, discussed the module development.

### 2.3. PHASE II

#### 2.3.1. Study Design

##### Content Validity

Huffaz ProHealth 1.0^©^ Module was developed, and its content validity was tested. A series of discussions between the researchers and Tahfiz school teachers and administrators were held to evaluate the module. The content validity evaluation was carried out by field experts and professionals.

The content validity assessment involved two parts: content validity for the item and overall acceptance of the component. A panel of experts were required to evaluate the validity of the module’s content by providing feedback on the developed questionnaires. Scoring on the questionnaire was based on four-point Likert scale (0 = strongly disagree, 1 = disagree, 2 = agree, 3 = strongly agree). The content validity of each item was evaluated using a content validity index. Items with I–CVI scores exceeding 0.78 were considered good and accepted by the panel of experts [10]. The assessment of acceptance as a whole was evaluated using the content validity value as adapted from Koo et al., (2018) and Clayton (2009) via the Tool to Evaluate Materials Used in Patient Education (TEMPtED) [9,11]. The criteria assessed included aspects of content, principles of motivation, literacy, layout and typography, and graphics. Content validity values of more than 70% for each criterion were considered appropriate and accepted by the panel [9,12].

##### Feasibility

The expected feasibility assessment is an analysis to assess whether a study is feasible and easy to conduct in a selected place or organization, including its technical and economic aspects [13]. The evaluation form of the developed questionnaire in this study included questions that were rated through a four-point Likert scale (0 = strongly disagree, 1 = disagree, 2 = agree, 3 = strongly agree). A mean score of 2 (agree) and above indicated that a component was accepted by the panel of experts and was suitable to be run in an organization [14].

#### 2.3.2. Study Location

The study was conducted among Tahfiz students in their schools that were registered under the Selangor Tahfiz Institutions Association (PITAS) located in Selangor Malaysia. The suggested data analysis method is as follows (Table 1):

## 3. Results

### 3.1. Phase I

The following results show the preliminary findings of Phase I of the study. Table 2 shows the demographic data of the needs assessment, consisting of male Tahfiz students and teachers, 20 and 12 of them, respectively.

Table 3 shows the needs assessment findings according to the physical activity, healthy diet, and psychological well-being aspects. Questions about the practice of physical activity and exercise, and the time spent exercising in a week (hours), were not answered by the teachers because they were meant to be explicitly answered by the students.

### 3.2. Phase II

#### 3.2.1. Content Validity

Table 4 shows the demographic data of the experts and teachers involved in the Phase II.

Meanwhile, Table 5 shows the content validity index for each item of each component in the Huffaz ProHealth 1.0^©^ module and Table 6 shows the result of the module criterion assessment.

#### 3.2.2. Feasibility Expectations

Table 7 shows the results of the assessment of feasibility expectations by experts.

## 4. Discussion

The Sidek Module Development Model [12] is often used as a reference to develop modules in Malaysia [16]. Three principal components in planning and implementing an effective program (module) are context, purpose, and operation. A module must have a theoretical framework to achieve development objectives [18]. Meanwhile, the crucial elements to ensure a module is useful are the content, the theory applied, the teaching design, the material used, and the material’s formative evaluation [19].

These thirteen steps are frequently used to develop motivational modules, teaching modules, training modules, and academic modules—for example, the psychospiritual relaxation therapy intervention module and stress management module among university students [16]. The Sidek Module Development Model was chosen as a reference due to its integrated and comprehensive model [20]. The module must go through an evaluation phase to identify existing weaknesses and shortcomings to ensure that it is of high quality and able to be implemented. A previously developed module using this model was a self-motivation module for high school students [21], which contained twelve activities resulting from the first phase, and obtained validity values and reliability values of 0.75 and 0.96, respectively. Besides that, other modules included a mind map-based teaching module for mathematics subject for high school students, an academic module [19], and an Islamic Spiritual Counseling (i-SC) module [20] for students’ behavior.

According to the World Health Organization (WHO), a health intervention is a planned act or strategy undertaken with or on behalf of a person or a population to assess, improve, promote, or modify human health, function, or health status. Health intervention modules are evaluations and actions that are systematically planned to be implemented on humans with such goals. A quality module leads a person towards positive change [22]. Innovative ideas are really needed to enhance healthy physical activity among children and adolescents [23].

Even though the health-related intervention modules are common in Malaysia, the information on the interventions is still lacking and limited. For example, a health-based physical fitness intervention program consisting of four additional physical exercises has been successfully executed in the form of circuit training (shuttle training, push-up exercises followed by jumps, star jumping exercises, and push-ups) among 86 sixteen-year-old male students. The exercises were inserted in their 40 min of Physical Education classes conducted twice a week for ten consecutive weeks. It successfully increased the students’ flexibility level. Flexibility is one of the health-based fitness components [24].

The physical health and fitness of adolescents depend on their daily physical activities and lifestyle. Adolescents who have a good and active lifestyle have a high level of physical health and fitness [25,26]. Good personal hygiene is very important, as it not only protects us from getting illnesses, but also improves our self-esteem. The development of the physical health component in this module could be seen as an effort to help Tahfiz students live more vibrant and active lives by encouraging them to be involved in hygiene practices and healthy physical activities.

This component’s personal hygiene unit consists of demonstrations of proper handwashing practices and the “Clean Hands Tahfiz” game. The environmental hygiene unit consists of talks and *gotong-royong* activities to raise their awareness towards environmental hygiene practices. Approaches involving teachers and students effectively improve hand hygiene practices among students [27]. *Gotong-royong* is used as a routine activity for students to maintain the school environment’s cleanliness while ensuring comfort for the learning process.

Physical fitness, which enables one to work effectively, enjoy leisure time, have enough rest time, fight diseases, and face emergencies should be achieved by all individuals regardless of gender and age. Individuals in the age group of 5 to 17 are recommended to perform moderate or high-level physical activity for at least 60 min daily [28]. High-level physical activity can be performed according to the individual’s ability.

The needs assessment of physical activity among students found that most students correctly understood physical health, and acknowledged the benefits of good physical activity to their health and memorization. However, only half of the students practiced exercises in daily life. This result was in line with the National Health and Morbidity Survey (NHMS), in which 36,2% adolescents aged 15–19 were physically inactive [29]. Among the causes of low level of physical activity by Tahfiz students were inadequate sports equipment and a lack of suitable space to do sports at their schools, thereby weakening the students’ motivation to stay active. In addition, schools also did not allocate a specific time slot for students to exercise, making them prefer to relax and do other activities to fill their free time [30].

The majority of the teachers showed a good understanding of physical health and awareness of the benefits of exercise to students’ health and memorization. The research needs assessment showed that the performance, attitude, and interests of Tahfiz school students towards healthy physical activities, such as sports and exercise, were at a low level, which was different from the results of needs assessment from the teachers. This result might be because the teachers have been exposed to broader and earlier knowledge and awareness of healthcare. No specific theories and modules could be referred to for physical health, since such modules had never been fully developed and implemented in Malaysia. Researchers had to refer to various types of modules and studies with different populations or study designs.

Students stated that sports and exercises affect their memorization level because such activities could relieve stress, keep the body healthy, and refresh the brain. For students who did not like sports and exercise, such activities were negligent and reduced their time memorizing the Quran. They believed that the effects of nutrition on the brain were more influential than physical activities. Meanwhile, all teachers agreed that sports and exercise were necessary for their students because they made the body sweat, refreshed the body, maintained fitness, reduced stress, and followed the practice of the Prophet Muhammad. A healthy and fit body could improve intelligence, memorization, and mental health [31]. However, time constraints and tight schedules became obstacles for the students.

The research needs showed that Tahfiz students had a good understanding of a healthy diet and awareness of its importance on body health and its effects on memorization. However, less than half of them were maintaining a healthy diet in their daily lives. Most of the students still lacked knowledge about nutrition and the concept of a healthy diet. The students had difficulty following practices or recommendations for a healthy diet because they were easily exposed to unhealthy diets, such as consuming snacks and fast food [32].

The eating patterns of Tahfiz students showed that only protein and sodium intake exceeded the Recommended Nutrient Intake percentage (% RNI) [30]; the other intakes, such as energy, carbohydrates, total fat, fiber, potassium, vitamin C, vitamin E, and zinc were less than the daily recommendations. Besides that, the nutritional deficiencies, especially vitamin A and vitamin C among students, were associated with a lack of vegetable intake [33]. The consumption of fruits, vegetables, whole grains, and low-fat and non-fat dairy products was much lower than it should be. The consequences were deficiencies in some essential nutrients, such as calcium, potassium, fiber, magnesium, and vitamin E.

Most students often did not take breakfast [34]. The habit of often leaving breakfast is very worrying because it can negatively impact students in terms of problem-solving skills, physical endurance, creativity, and well-being [35]. On the other hand, students who have breakfast show better academic performance and emotional function [36,37]. In contrast, the energy intake from solid fat sources and added sugars was found to far exceed the recommendations [38].

The surveys conducted among teachers showed a good understanding of a healthy diet. They were aware of its importance to their students, believing that a healthy diet could affect students’ memorization. There was also conflicting result among the teachers pertaining to the practice of healthy diet in their daily life. This result shows that teachers and Tahfiz schools need to work together to improve the teaching methods or implement healthy diets for students, as teachers play an important role in modelling positive and healthy eating behaviors [39]. Learning about nutrition could increase the awareness among students regarding the importance of nutritional education [40].

Since boarding school students only consume food provided by the school canteen, it is essential for the school to create an environment that promotes a healthy diet. These schools have an influence on the food selection and quality of food chosen by students every day. This healthy diet environment could be achieved by ensuring that the canteen operators serve healthy and nutritious food in accordance with the food pyramid guide [34].

The interpersonal relationships, such as the students’ relationships with their parents, teachers, and people around them, impact their emotion and memorization. Moral support from close individuals significantly helps control the student’s emotions [41]. Parents and teachers need to educate students on how to control their emotions, especially during stressful situations. Good emotions would enhance al-Quran memorization and studies, whereas emotional problems would lead to stress and distract their focus.

Briefly, there were four activities in the psychological well-being component. Motivational seminars and time and stress management seminars were aimed to provide a basic introduction and exposure. Periodic video shows were aimed to inspire students in their daily lives through lessons learned from the shown videos. An interactive discussion and learning methods were implemented through Five Minute Motivation (MOT5) activities and Peers of Tahfiz (PRT). Other methods, such as the Rewards and Fines System, Expression Box, and Counselling on Request/Referral were also designed to make sure that this module would be useful and fitting the target population.

In this study, all units for physical health components, namely, personal hygiene, environmental hygiene, and physical activity, showed high I–CVI values in the range of 95.84% to 100%, exceeding the 78% threshold. Therefore, the physical health component had good content validity. There were several suggestions for improvement given by the panel of evaluators. The experts recommended to explain hand-washing in detail to ensure that teachers can convey information related to proper hand-washing techniques to their students. As for stretching exercises, experts suggested adding specific objectives and stating the appropriate time to do each type of stretching exercise.

The nutrition component recorded an I–CVI value of 100% for the three units, namely, the quarter-quarter-half concept (QQHC), food pyramid, and healthy menu planning; and 91.67% for the food hygiene and safety units. All units had I–CVI values exceeding 78%, proving that the nutritional component had good content validity. Experts suggested that teachers need to be empowered with solid basic knowledge of nutrition, especially involving module activities such as the QQHC and the food pyramid. In addition, the students must be involved in the activities in food hygiene and safety, instead of canteen operators and school teachers only.

The psychological well-being component also had I–CVI values of more than 78% for all units, namely, the self-motivation unit and time and stress management unit. The range of I–CVI values recorded was from 85.42% to 97.92%. The time and stress management unit had a relatively low I–CVI value because the Tahfiz peer mentor activity only had an I–CVI value of 75%, indicating that it did not achieve good content validity.

The second part of the content validity questionnaire was on the criteria evaluation section of the module (TEMPtED). The evaluation was done based on five criteria: content, principles of motivation, literacy, layout, and typography and graphics.

The content validity of the physical health component was categorized as accepted for each criterion which was in the range of 78–85%. Among the suggestions given by the panels to improve understanding was to provide a neater component introduction by breaking it into several paragraphs. For the personal hygiene unit, they suggested a video on how to wash hands properly. They also suggested providing more effective and interesting posters by using animations and cartoon strips to better engage the students.

For the environmental hygiene unit, the panels stated that the template provided for the speakers was good. However, for the exhibition activities, more attractive posters, for instance, in the form of comics, should be provided. For the physical activity unit, they suggested that the picture of the deep breathing exercise guide should be clearer. As for the correct body posture, the experts recommended improving the picture for each correct body posture guide by placing appropriate angles for each position for better guidance. For eye exercise activities, the poster size needs to be enlarged. As for the muscle resistance exercise activities, there was a suggestion to convert the instructions to active sentences. There was a suggestion to add “please refer to the video” note in the module text to ensure that teachers were aware of the video provided.

The content validity of nutritional component showed more than 70% for each criterion except for the layout and graphic criterion, which only reached a score of 67%. Therefore, all criteria were considered acceptable except the layout and graphics criterion due to the irregular arrangement of units and activities. The experts suggested that the order of the positions of units 2.3 and 2.4 to be reversed, and that the basics of food safety in unit 2.3 should be rearranged.

Meanwhile, for the psychological well-being component, the content validity scored more than 70%, except the content criterion, which had a score of 60%. This is because there was unclear content and instructions for Tahfiz peer mentor activities.

The third part of the content validity questionnaire was the expected feasibility. A total of 58.33% of experts strongly agreed and 41.67% of experts agreed that the physical health component could be implemented; the average score given by the experts was 2.58, and the range of scores was two in size. Meanwhile, for the nutrition component, 58.33% of experts strongly agreed and 41.67% of experts agreed that this component could be implemented; the average score given by experts was 2.58, and the range of scores was two in size. For the psychological well -being component, 66.67% of experts strongly agreed and 33.33% of experts agreed that the psychological well-being component could be implemented; the average score given by experts was 2.67, and the range of scores was two in size.

This study has several limitations and challenges. The needs assessment only involved male students aged 13–17 years old. The daily schedules and curricula of Tahfiz school students are very packed. Therefore, there were difficulties in adjusting activities in this study according to the needs, daily schedules, and facilities available in Tahfiz schools. The content validity was done with a relatively small number of experts. As this module is meant for an educational purpose, the content validity shall be determined with a larger sample size.

## 5. Conclusions

In this study, the approaches and methods for developing and evaluating the Huffaz ProHealth 1.0^©^ Module were explored. The needs assessment and module development were successfully executed. Based on the current validity assessment for all components, this module is suitable to be employed at Tahfiz schools.

## Figures and Tables

**Figure 1 ijerph-19-07718-f001:**
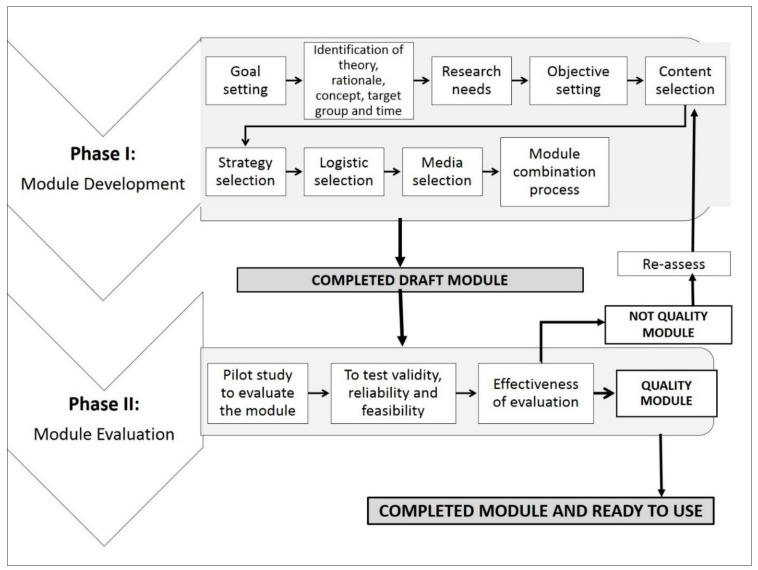
Sidek Module Development Model.

**Table 1 ijerph-19-07718-t001:** Content validity calculation and effectiveness statistical analysis.

Part	Calculation Method	Formula/Statistical Analysis	Reference
I	Item-level Content Validity Index (I–CVI)	$I-CVI=\frac{agreed\;item}{total\;number\;of\;experts}$	Polit et al., (2007) [10], Norlia & Faizah (2016) [15]
II	Content Validity Level	$Content\;Validity\;Level=\frac{total\;expert\;score}{total\;maximum\;score}\times 100$	Sidek & Jamaludin (2005) [12], Clayton (2009) [9], Mat Rasik & Ismail (2019) [16], Zaharen, Saper & Nasir (2019) [17]
III	Effectiveness of Module	Generalized Linear Model-Mixed Design ANOVA	-

**Table 2 ijerph-19-07718-t002:** Demographic data of Tahfiz school students and Tahfiz school teachers.

Demographic Data of Tahfiz School Students		
Demographic Data	Category	Frequency, *n* (%)
Age (years)	12	1 (5 %)
	13	4 (20%)
	14	3 (15%)
	15	9 (45%)
	16	2 (10%)
	17	1 (5%)
Tahfiz school	Maahad Tahfiz A	20 (100%)
Study period (years)	0–3	15 (75%)
	4–6	4 (20%)
	>6	1 (5%)
Level of Al-Quran memorization (juzuk)	0–10	12 (60%)
	11–20	7 (35%)
	12–30	1 (5%)
**Demographic Data of Tahfiz School Teachers**		
**Demographic Data**	**Category**	**Frequency, *n* (%)**
Age (years)	20–29	7 (58.33%)
	30–39	4 (33.33%)
	40–49	1 (8.33%)
Tahfiz school	Maahad Tahfiz B	6 (50%)
	Maahad Tahfiz C	6 (50%)
Teaching experience (years)	0–3	5 (41.67%)
	4–6	4 (33.33%)
	7–9	2 (16.67%)
	>9	1 (8.33%)
Education level	Secondary/religious school	5 (41.67%)
	Diploma/STPM/Certificates	4 (33.33%)
	Degree	2 (16.67%)
	Others	1 (8.33%)

**Table 3 ijerph-19-07718-t003:** Tahfiz school students and teachers’ responses to the physical health, healthy diet, and psychological well-being questions.

Tahfiz School Students and Teacher’s Responses on the Physical Health Aspect				
Item	Teachers		Students	
	Response	*n* (%)	Response	*n* (%)
Understanding of physical health	Healthy, no chronic diseases Active and energetic Healthy, fit and focus while studying Healthy, high immunity Physically and mentally healthy Not sure	2 (16.7%) 3 (25.0%) 2 (16.7%) 1 (8.33%) 2 (16.7%) 2 (16.7%)	Active/healthy lifestyle practice Healthy body Healthy brain Healthy eating Help in learning Not sure	9 (45.0%) 4 (20.0%) 3 (15.0%) 2 (10.0%) 1 (5.0%) 1 (5.0%)
Exercise is important for health	Yes	12 (100%)	Yes No Not sure	17 (85.0%) 1 (5.0%) 2 (10.0%)
Benefits of healthy physical activity	Body sweating Enhance brain memorization Refresh and nourish the body Make wake up early in the morning easier Increase fitness and focus Reduce stress Physically and mentally healthy Important	2 (16.7%) 3 (25.0%) 2 (16.7%) 1 (8.3%) 1 (8.3%) 1 (8.3%) 1 (8.3%) 1 (8.3%)	Prevent diseases Sharpen the brain Enhance the Quran memorization Make the body healthy and fit Body sweating Supplying energy Not sure	1 (5.0%) 2 (10.0%) 1 (5.0%) 6 (30.0%) 3 (15.0%) 1 (5.0%) 6 (30.0%)
Practice on physical activity and exercise			Yes No Not sure	10 (50.0%) 8 (40.0%) 2 (10.0%)
Time spent for exercise in a week (hours)			<1 1–2 2–3 >3	4 (20.0%) 10 (50.0%) 5 (25.0%) 1 (5.0%)
Exercise affect Al-Quran memorization	Yes	12 (100%)	Yes No Not sure	12 (60.0%) 4 (20.0%) 4 (20.0%)
Effect of exercise on Al-Quran memorization?	Maintain body fitness Enhance intelligence Enhance memorization capacity Reduce stress Not affected. excessive exercise makes students lazy Not sure	3 (25.0%) 4 (33.3%) 1 (8.3%) 2 (16.7%) 1 (8.3%) 1 (8.3%)	Make the body healthy and fit Sharpen the brain Reduce stress Enhance the Quran memorization Not affected. Exercise is just tiring Not affected. The brain is only affected by nutrition Not sure	4 (20.0%) 6 (30.0%) 1 (5.0%) 2 (10.0%) 1 (5.0%) 1 (5.0%) 5 (25.0%)
**Tahfiz School Students and Teacher’s Responses on the Healthy Diet Aspect**				
**Item**	**Teachers**	**Students**		
	**Response**	***n* (%)**	**Response**	***n* (%)**
Understanding of healthy diet	Nutritious Halal and clean Guarantee health A diet that follows the teaching of the Prophet	5 (41.7%) 2 (16.7%) 4 (33.3%) 1 (8.3%)	Helps in memorization Nutritious Healthy body Complete nutrition Healthy mind Do not know/Not sure	3 (15.0%) 4 (20.0%) 2 (10.0%) 5 (25.0%) 1 (5.0%) 5 (25.0%)
A healthy diet is essential for health, learning and memorization	Agree	12 (100%)	Agree Disagree Not sure	18 (90.0%) 0 (0.0%) 2 (10.0%)
Selection and consumption of a healthy diet affect the performance of memorization	Yes	12 (100%)	Yes No Not sure	18 (90.0%) 1 (5.0%) 1 (5.0%)
Effects of a healthy diet on the performance of memorization	Improve memorization performance Emphasis on the concept of “halalan toyyiban” To prevent dangerous diseases	8 (66.7%) 3 (25.0%) 1 (8.3%)	Helps in memorization Produce healthy body Create healthy mind Don’t know/Not sure	8 (40.0%) 2 (10.0%) 3 (15.0%) 7 (35.0%)
Practice on a healthy diet	Yes No Not sure	5 (41.7%) 3 (25.0%) 4 (33.3%)	Yes No Not sure	4 (20.0%) 9 (45.0%) 7 (35.0%)
**Tahfiz School Students and Teacher’s Responses on the Psychological Well-Being Aspect**				
**Item**	**Teachers**		**Students**	
	**Response**	***n* (%)**	**Response**	***n* (%)**
Emotion affects Students’ Memorization	Yes No Not sure	12 (100%) 0 (0%) 0 (0%)	Yes No Not sure	13 (65.0%) 2 (10.0%) 5 (25.0%)
Effects of Emotion on Students’ Memorization	Difficult memorization affects emotion Anxiety disturbs memorization Calmness important in memorization Stress affects memorization	2 (16.7%) 3 (25.0%) 4 (33.3%) 3 (25.0%)	Emotion would affect memorization Sad emotion disturbs memorization Stress can affect memorization Calmness important in memorization Others Not sure	7 (35.0%) 2 (10.0%) 1 (5.0%) 1 (5.0%) 2 (10.0%) 7 (35.0%)
Students’ emotion was affected by relationships with parents, teachers and others	Yes No Not sure	12 (100%) 0 (0%) 0 (0%)	Yes No Not sure	16 (80.0%) 2 (10.0%) 2 (10.0%)
Effects of relationships with parents, teachers and others on students’ emotion	Good relationship stabilizes emotion Support and a positive environment reduce discipline problem and increase focus A bad relationship affects emotions and focus Parents and teachers help in building positive emotions	5 (41.7%) 1 (8.3%) 2 (16.7%) 4 (33.3%)	A good relationship leads to success A good relationship ensures peacefulness Not sure	11 (55%) 4 (20.0%) 5 (25.0%)
Students’ frequency of feeling sad and angry	Never Rarely Sometimes Too often All the time	2 (16.7%) 2 (16.7%) 8 (66.7%) 0 (0%) 0 (0%)	Never Rarely Sometimes Too often All the time	5 (25.0%) 4 (20.0%) 8 (40.0%) 2 (10.0%) 1 (5.0%)
Actions taken by students when sad or angry	Express to teachers/friends Crying Adding spiritual practice Being alone/depression Angry/Breaking the rules Not focused on memorizing	4 (33.3%) 1 (8.3%) 2 (16.7%) 3 (25.0%) 1 (8.3%) 1 (8.3%)	Adding spiritual practice Do enjoyable activities Crying Sleep Self-hitting Hiding feelings/loneliness Not sure	6 (30.0%) 3 (15.0%) 1 (5.0%) 1 (5.0%) 1 (5.0%) 6 (30.0%) 2 (10.0%)

**Table 4 ijerph-19-07718-t004:** Demographic data.

Demographic Data	Experts (*n* = 5)	Teachers (*n* = 7)
Sex		
Male	1 (20%)	6 (85.7%)
Female	4 (80%)	1 (14.3%)
Education backgrounds		
Diploma	-	2 (28.6%)
Bachelor’s degree	-	3 (42.9%)
Master’s degree	-	1 (14.3%)
Doctor of Philosophy	5 (100%)	1 (14.3%)

**Table 5 ijerph-19-07718-t005:** Content validity index for each item of each component in the Huffaz ProHealth 1.0^©^ module.

Component	I–CVI	Acceptance
**Physical health**		
Deep breathing exercises	0.92	Accepted
Correct body posture	1.00	Accepted
Stretching exercises	0.92	Accepted
Eye exercises	1.00	Accepted
Muscle resistance exercises	1.00	Accepted
**Nutrition**		
Quarter quarter half concept	1.00	Accepted
Food pyramid	1.00	Accepted
Food hygiene and safety	1.00	Accepted
Healthy menu planning	0.92	Accepted
**Psychological well-being**		
Time and stress management seminars	0.92	Accepted
Tahfiz peer mentor	0.75	Accepted
Feelings expression box	0.92	Accepted
Counseling on demand	0.83	Accepted

**Table 6 ijerph-19-07718-t006:** Module criterion assessment (TEMPtED).

Component	Mean ± SD	CVI (%)	Acceptance
**Physical health**			
Content	21.33 ± 3.60	79.00	Accepted
Principle of motivation	4.75 ± 1.21	79.00	Accepted
Literacy	9.83 ± 1.70	81.94	Accepted
Layout and typography	12.75 ± 2.22	85.00	Accepted
Graphic	7.75 ± 1.42	86.11	Accepted
**Nutrition**			
Content	22.58 ± 3.87	83.64	Accepted
Principle of motivation	4.58 ± 0.90	76.39	Accepted
Literacy	10.00 ± 1.86	83.33	Accepted
Layout and typography	10.08 ± 1.73	67.22	Not accepted
Graphic	7.83 ± 1.40	87.04	Accepted
**Psychological well-being**			
Content	16.50 ± 3.12	61.11	Not accepted
Principle of motivation	4.33 ± 1.23	72.22	Accepted
Literacy	10.50 ± 1.88	87.50	Accepted
Layout and typography	12.17 ± 2.29	81.11	Accepted
Graphic	7.42 ± 1.50	82.41	Accepted

**Table 7 ijerph-19-07718-t007:** Assessment of feasibility expectations by experts.

Component	Agree (%)	Strongly Agree (%)	Mean ± SD	Acceptance
Physical health	41.67	58.33	2.58 ± 0.51	Accepted
Nutrition	41.67	58.33	2.58 ± 0.51	Accepted
Psychological well-being	33.33	66.67	2.67 ± 0.51	Accepted

## Data Availability

The data presented in this study are available on request from the corresponding author.

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
