# Peer review of "Development and Validation of Huffaz ProHealth 1.0©: A Module to Improve the Well-Being of Tahfiz Students in Selangor, Malaysia"

_ijerph, 2022, doi:10.3390/ijerph19137718_

Round 1

Reviewer 1 Report

The topic addressed is interesting. Multi-disciplinary approaches were used to develop the module as a comprehensive module towards psychological and physical well-being and as a guide to improve students' health and quality of life. The study design proposes to conduct an extended experiment used to assess the validity, reliability, and feasibility of the module, and evaluation indicators were explicitly selected. Results are described clearly. The article could be improved and enhanced in some areas, and I have outlined my most prominent concerns below.

  1. In the introduction, you need to relate the cited literature to the objectives of the paper and provide a concise analysis of the current situation based on the literature review. Consider whether the "media selection" section of the introduction and the " be evaluated based on five areas" in line 95 are relevant to the purpose of the study, as they are not reflected in the subsequent study.
  2. In 2.2, "validity, reliability, and feasibility" are mentioned, but in 2.3, only "validity and reliability" are described. Would the authors possibly consider describing how to conduct "feasibility testing" in 2.3?
  3. Please clarify the duration of the extended experiment, the measurement intervals mentioned in lines 187 and 208 of the article are three months and two months respectively.
  4. I wonder if you could clarify which survey method was used for the "level of food intake" in line 184?
  5. The discussion section is the main part of the study. Authors cite a lot of literature to make the discussion more comprehensive. Authors can streamline the content to make the findings and contributions of the study clearer.
  6. In line 334, "all teachers showed a good understanding of physical health", there is no data to support this conclusion, as 16.7% of teachers selected "not sure" in the relevant item in Table 3.
  7. In line 358, "only a few of them were practicing a healthy diet", under the item "Practice on a healthy diet" in Table 4, the results of the survey differed between teachers and students, and the above conclusion could not be drawn clearly.
  8. In line 378, "students did not practice a healthy diet", but in Table 4, 25% of the teachers thought that students did not practice a healthy diet in their daily life, which is not the opinion of all teachers. In the conclusion, it is recommended to use quantitative reasoning in conjunction with the results and to make comparisons based on the combined literature.

   Overall, I do think that the topic is very interesting and I wish the authors the best of luck on their further endeavors.

Author Response

1.

In the introduction, you need to relate the cited literature to the objectives of the paper and provide a concise analysis of the current situation based on the literature review. Consider whether the "media selection" section of the introduction and the " be evaluated based on five areas" in line 95 are relevant to the purpose of the study, as they are not reflected in the subsequent study.

After a major revamp of this manuscript, we have removed the mentioned sections.

2.

In 2.2, "validity, reliability, and feasibility" are mentioned, but in 2.3, only "validity and reliability" are described. Would the authors possibly consider describing how to conduct "feasibility testing" in 2.3?

We have revised the manuscript and only report the content validity and feasibility.

(line160 – 188)

3.

Please clarify the duration of the extended experiment, the measurement intervals mentioned in lines 187 and 208 of the article are three months and two months respectively.

We have removed this part

4.

I wonder if you could clarify which survey method was used for the "level of food intake" in line 184?

We have removed this part

5.

The discussion section is the main part of the study. Authors cite a lot of literature to make the discussion more comprehensive. Authors can streamline the content to make the findings and contributions of the study clearer.

We have revised according to the new focus of this manuscript.

6.

In line 334, "all teachers showed a good understanding of physical health", there is no data to support this conclusion, as 16.7% of teachers selected "not sure" in the relevant item in Table 3.

We revised it into
Majority of the teachers showed .. .. (Line 294)

7.

In line 358, "only a few of them were practicing a healthy diet", under the item "Practice on a healthy diet" in Table 4, the results of the survey differed between teachers and students, and the above conclusion could not be drawn clearly.

Revised into

However, less than half of them were practicing a healthy diet .. (Line315)

8.

In line 378, "students did not practice a healthy diet", but in Table 4, 25% of the teachers thought that students did not practice a healthy diet in their daily life, which is not the opinion of all teachers. In the conclusion, it is recommended to use quantitative reasoning in conjunction with the results and to make comparisons based on the combined literature.

We have revised into

There was also conflicting result among the teachers pertaining to practice of healthy diet in their daily life. (Line 337-338)

Reviewer 2 Report

This manuscript described the methods of developing an intervention program to improve psychological and physical well-being among Tahfiz students in Malaysia.

The current manuscript was a report on the methods of developing Phase I and Phase II of an intervention program as well as the result of Phase I. It is not a report on the results of a study with clear study aim and rigorous study design. Most of the contents in the present manuscript described the steps to develop the draft module but not the evidence to support the results of Phase I. Therefore, the current content of this manuscript did not reach a standard to be publish as a original article.

Author Response

1.

This manuscript described the methods of developing an intervention program to improve psychological and physical well-being among Tahfiz students in Malaysia.

The current manuscript was a report on the methods of developing Phase I and Phase II of an intervention program as well as the result of Phase I. It is not a report on the results of a study with clear study aim and rigorous study design. Most of the contents in the present manuscript described the steps to develop the draft module but not the evidence to support the results of Phase I. Therefore, the current content of this manuscript did not reach a standard to be publish as a original article.

We have made a substantial revision on the content of this manuscript. We focus on the module development alone. The protocol of the overall study will be in another manuscript.

Reviewer 3 Report

I think it's worthwhile to develop a health module. It was well planned for evaluating the validity of the module developed for practical application. If you increase the readability of the manuscript, it can provide useful information for future research or clinical application. I evaluate it as a manuscript that can be published in this journal if somethings are supplemented. Those things are as follows.

  1. The title of the manuscript is not specific. Based on what you study, the title doesn't seem appropriate.

  1. I think it would be good to summarize the abstract in 2/3 of what it is now.

  1. Please describe the subheading in more detail.

  1. Shouldn't the content validity be targeted at health professionals or medical personnel? If you target teachers or students, it's face validity.

  1. The content validity of a scale or treatment program that can be statistically verified in the future can be examined with a small sample (with few experts), but the content validity of an educational program or module that should be examined with a larger sample size. I think this is a weakness of this study, so it should be presented as a limitation of the study.

  1. Perhaps because of the above problem, the interpretation of the research results is very limited.

  1. In this study, there should be a criterion for CVI, and it should provide the basis for how it was determined.

  1. Due to the composition of the manuscript, it is less readable when the reader reads this manuscript, so it needs to be supplemented.

Author Response

1.

The title of the manuscript is not specific. Based on what you study, the title doesn't seem appropriate.

Major revision has been made and a new title is

Development and Validation of Huffaz Prohealth 1.0©: A Module to Improve the Well-Being of Tahfiz Students in Selangor, Malaysia.

2.

I think it would be good to summarize the abstract in 2/3 of what it is now.

Abstract has been revised according to new direction of this manuscript.

3.

Please describe the subheading in more detail.

We have revised this manuscript thoroughly

4.

Shouldn't the content validity be targeted at health professionals or medical personnel? If you target teachers or students, it's face validity.

We include both professionals and the field (teachers) experts

5.

The content validity of a scale or treatment program that can be statistically verified in the future can be examined with a small sample (with few experts), but the content validity of an educational program or module that should be examined with a larger sample size. I think this is a weakness of this study, so it should be presented as a limitation of the study.

Thank you pointing this out. We have added in the study limitation

The content validity was done on a relatively small number of experts. As this module is meant for an educational purpose, the content validity shall be determined with a larger sample size. 

6.

Perhaps because of the above problem, the interpretation of the research results is very limited.

We acknowledge this and revised our conclusion to avoid overclaiming of the findings.

Meanwhile, based on the current validity assessment for all components, indicated that this module is suitable to be employed at Tahfiz schools.

7.

In this study, there should be a criterion for CVI, and it should provide the basis for how it was determined.

We explained in the method section

As for the assessment the acceptance as a whole, it was evaluated using the content valid-ity value that has been adapted from Koo et al. (2018) and Clayton (2009) through the Tool to Evaluate Materials Used in Patient Education (TEMPtED). The criteria assessed in-cluded aspects of content, principles of motivation, literacy, layout and typography and graphics. Content validity values more than 70% for each criterion are considered appro-priate and accepted by the expert panel (Clayton 2009; Mohd & Ahmad 2005). .. (Line 173-178)

8.

Due to the composition of the manuscript, it is less readable when the reader reads this manuscript, so it needs to be supplemented.

We have revised the manuscript thoroughly and presented only the module development part.

Round 2

Reviewer 2 Report

The current manuscript was a report on the methods of developing Phase I and Phase II of an intervention program as well as the result of Phase I. It is not a report on the results of a study with clear study aim and rigorous study design. Most of the contents in the present manuscript described the steps to develop the draft module but not the evidence to support the results of Phase I. Therefore, the current content of this manuscript did not reach a standard to be publish as a original article.

Author Response

We submitted our revised manuscript with a significant changes. We had removed the methodology for the whole study  and kept only the module development.

We explained 2 Phases of our module development:
1. Phase 1 - need Assessment & Development
2. Phase 2 - validation  

We reported the findings for the need assessment  in the Result section (Line 189 - Line 203)

We reported the finding for the validation and feasibility (Line 205 - Line 224)

There are similar  studies reporting the module development (need assessment and validation) processes that have been published before. For the matter of fact, IJEPRH itself has published similar articles before. The article structures are similar to our. They can found below:

1. Lau, X.C.; Wong, Y.L.; Wong, J.E.; Koh, D.; Sedek, R.; Jamil, A.T.; Ng, A.L.O.; Hazizi, A.S.; Ruzita, A.T.; Poh, B.K. Development and Validation of a Physical Activity Educational Module for Overweight and Obese Adolescents: CERGAS Programme. Int. J. Environ. Res. Public Health 2019, 16, 1506. https://doi.org/10.3390/ijerph16091506

2. Kasim, N. H., & Ahmad, C. N. C. (2018). PRO-STEM Module: The Development and Validation. International Journal of Academic Research in Business and Social Sciences, 8(1), 728–739.

3. Ignjatovic A, Thomas-Gibson, S., East, JE, Haycock A., Bassett, P., PBhandari, P., Man, R.,  Suzuki, N. & Saunders, B.P. 2011. Development and validation of a training module on the use of narrow-band imaging in differentiation of small adenomas from hyperplastic colorectal polyps. Gastrointestinal Endoscopy,73(1): 128-133

4. Klassen, A.F., Dominici, L., Fuzesi, S. et al. Development and Validation of the BREAST-Q Breast-Conserving Therapy Module. Ann Surg Oncol 27, 2238–2247 (2020). https://doi.org/10.1245/s10434-019-08195-w

5. Kakde, Noopur, Metri, Kashinath G., Varambally, Shivarama, Nagaratna, Raghuram and Nagendra, H.R.. 2017. Development and validation of a yoga module for Parkinson disease. Journal of Complementary and Integrative Medicine 14(3): 20150112. https://doi.org/10.1515/jcim-2015-0112